**Subject Area:**
biotechnology/cellular biology/molecular biology

polyunsaturated fatty acids (PUFAs), n-3/6 PUFAs, fatty acid desaturase (Fad), *Fad2/3*, transgenic mice

**Author for correspondence:**
Guangpeng Li
e-mail: gpengli@imu.edu.cn

†These authors have contributed equally to the study.

# Generation of *Fad2* and *Fad3* transgenic mice that produce n-6 and n-3 polyunsaturated fatty acids

Lishuang Song[1,2,†], Lei Yang[1,†], Jiapeng Wang[1,2], Xuefei Liu[1], Lige Bai[1,2], Anqi Di[1,2] and Guangpeng Li[1,2]

[1]State Key Laboratory of Reproductive Regulation and Breeding of Grassland Livestock (R2BGL), and [2]College of Life Science, Inner Mongolia University, Hohhot 010070, People's Republic of China

LY, 0000-0002-9767-9026; GL, 0000-0001-8318-5853

Linoleic acid (18 : 2, n-6) and α-linolenic acid (18 : 3, n-3) are polyunsaturated fatty acids (PUFAs), which are essential for mammalian health, development and growth. However, the majority of mammals, including humans, are incapable of synthesizing n-6 and n-3 PUFAs. Mammals must obtain n-6 and n-3 PUFAs from their diet. Fatty acid desaturase (Fad) plays a critical role in plant PUFA biosynthesis. Therefore, we generated plant-derived *Fad3* single and *Fad2–Fad3* double transgenic mice. Compared with wild-type mice, we found that PUFA levels were greatly increased in the single and double transgenic mice by measuring PUFA levels. Moreover, the concentration of n-6 and n-3 PUFAs in the *Fad2–Fad3* double transgenic mice were greater than in the *Fad3* single transgenic mice. These results demonstrate that the plant-derived *Fad2* and *Fad3* genes can be expressed in mammals. To clarify the mechanism for *Fad2* and *Fad3* genes in transgenic mice, we measured the PUFAs synthesis-related genes. Compared with wild-type mice, these *Fad* transgenic mice have their own n-3 and n-6 PUFAs biosynthetic pathways. Thus, we have established a simple and efficient method for *in vivo* synthesis of PUFAs.

## 1. Introduction

Polyunsaturated fatty acids (PUFAs) consist of more than two double bonds, and are required for mammalian and human health [1,2]. There are two main categories of PUFAs of n-6 PUFAs (also known as omega-6, ω-6) and n-3 PUFAs (or ω-3) [3]. The n-6 PUFAs mainly include linoleic acid (LA; 18 : 2, n-6), γ-linolenic acid (γ-LA; 18 : 3, n-6) and arachidonic acid (AA; 20 : 4, n-6). The n-3 PUFAs mainly include α-linolenic acid (ALA; 18 : 3, n-3), eicosapentaenoic acid (EPA; 20 : 5, n-3) and docosahexaenoic acid (DHA; 22 : 6, n-3) [1,4].

In a human newborn study, PUFAs are required to maintain normal retinal and brain development [5]. For children and adults, PUFAs can reduce the risk of some diseases such as vascular diseases, arthritis, cancer and neurological diseases [6–9]. Plants and microorganisms can synthesize PUFAs by themselves because they have their own desaturases [3,10]. However, mammals are unable to synthesize n-6 and n-3 PUFAs [11], because they lack delta (Δ)-12 and Δ-15 desaturases. Therefore, mammals and humans must consume the necessary n-6 and n-3 PUFAs in their diet.

Desaturation is an important biochemical process in fatty acid biosynthesis. Previous studies have shown that fatty acid desaturase (Fad) plays an important role in fatty acid desaturation [12,13]. The Fad enzymes convert saturated fatty acids with a single bond between two carbon atoms to unsaturated fatty acids with a double bond at a specific location in the fatty acyl chain [14,15]. The Fad2 enzyme introduces a double bond at the Δ12 position in the monounsaturated fatty acid oleic acid (OA; 18 : 1, n-9) to form n-6 LA, and the Fad3 enzyme

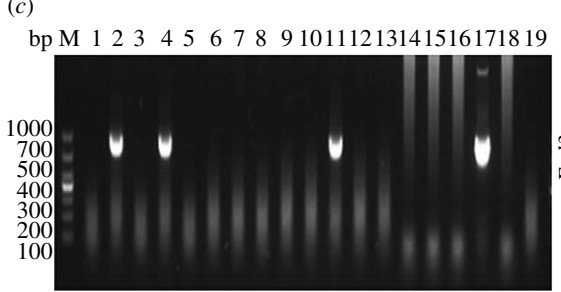

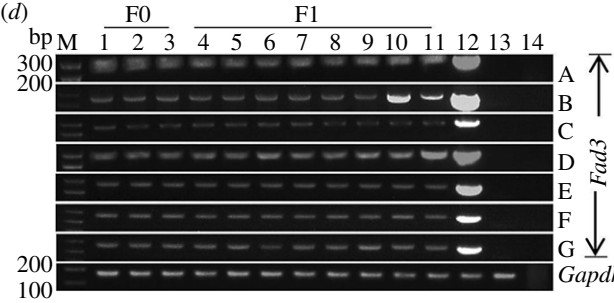

**Figure 1.** Generation of transgenic mice that express the *Fad3* gene. (*a*) Schematic representation of pCMV–Fad3 plasmid. (*b*) Summary of transgenic mice that carried the *Fad3* gene. (*c*) Identification of *Fad3* transgenic mice by PCR (detection of *Fad3* gene 865 bp; M: Marker DL1000; lanes 1–16: offspring mice; lane 17: positive control using wild-type genomic DNA mixed with pCMV–Fad3 plasmid as template; lane 18: negative control using wild-type genomic DNA as template; lane 19: blank control using H$_2$O as template). (*d*) Validation of *Fad3* transgene expression in seven major tissues by RT-PCR (the PCR product for *Fad3* and *Gapdh* cDNA is 235 bp and 193 bp, respectively; M: Marker DL1000; lanes 1–11: transgenic mice; lane 12: positive control using wild-type cDNA mixed with pCMV–Fad3 plasmid as template; lane 13: negative control using wild-type cDNA as template; lane 14: blank control using H$_2$O as template; A: skeletal muscle; B: fat; C: heart; D: liver; E: spleen; F: lung; G: kidney).

further converts n-6 LA to n-3 ALA by introducing a double bond in LA at the Δ15 position [11,16,17]. Unlike plant-derived Fad enzymes, another type of PUFAs synthetase *Fat1* (fatty acid metabolism 1) is derived from the roundworm *Caenorhabditis elegans* [13]. The *Fat1* was identified through its homology with the *Fad2/3* genes of *Arabidopsis*, and only converting n-6 to n-3 PUFAs [13].

Our previous studies have demonstrated that the *Fat1* transgenic livestock is capable of producing high levels of n-3 PUFAs in their tissues, organs and milk [18–23]. Although the content of n-3 PUFAs increased in *Fat1* transgenic animals, the content of n-6 PUFAs decreased. Previous studies have shown that n-6 PUFAs also play an important role in mammalian health, and the lack of n-6 PUFAs results in several diseases, such as monneuronal abnormalities, reduced growth, reproductive failure, skin lesions, fatty liver and polydipsia [6,17,24,25]. Furthermore, there may be public concern about the food safety of genetically modified animals made from *C. elegans*-derived genes.

The *Fad* desaturase genes have been successfully isolated from higher plants, including spinach and scarlet weeds [26,27]. In the present study, we generated *Fad3* single and *Fad2–Fad3* double transgenic mice to investigate whether plant-derived genes for a fatty acid desaturase can be functionally expressed in animals. We provide multiple lines of evidence showing functional expression of the *Fad2* and *Fad3* genes in the transgenic mice in major tissues, including skeletal muscle, fat, heart, liver, spleen, lung and kidney. Through analysis of PUFAs levels, we revealed that the *Fad3* single transgenic mice produce high levels of n-3 PUFAs from n-6 PUFAs. In *Fad2–Fad3* double transgenic mice, the *Fad2* desaturase promoted *de novo* synthesis of n-6 PUFAs, and the Fad3 desaturase further converted the n-6 into n-3 PUFAs. Therefore, the *Fad2–Fad3* double transgenic mice have established their own PUFA biosynthesis system.

# 2. Material and methods

## 2.1. Animals and chemicals

The CD1, Kun-Ming (KM) and BDF1 (C57BL/6N × DBA/2) mice were purchased from the Experimental Animal Research Center (Inner Mongolia University). Mice were housed in a specific pathogen-free room with an appropriate temperature (22–24°C) and light–dark cycle (light 8 .30–20.30), and were fed with a regular diet. All chemicals used in this study were purchased from Sigma (St Louis, MO, USA), unless otherwise indicated.

## 2.2. Expression vector construction of the *Fad2/3* gene

The *Fad2* and *Fad3* genes used in this study were PCR-amplified from the cDNA of *Spinacia oleracea* and *Linum usitatissimum* (also known as common flax or linseed), respectively. Total RNA was extracted from *S. oleracea* and *L. usitatissimum* using the RNeasy Plant Mini Kit (Qiagen, Hilden, Germany). The complete open reading frames and partial up- and downstream noncoding regions of the *Fad2/3* genes from cDNA were amplified using Phanta Super-Fidelity DNA Polymerase (Vazyme, Nanjing, China). The PCR products were cloned into the pMD19-T vector (Takara, Kusatsu, Japan) and sequenced according to the manufacturer's instructions. After sequencing, the *Fad3* cDNA was introduced into the pIRES expression vector at the *Xho*I and *Mlu*I restriction sites, and the vector was named pCMV–Fad3 (figure 1*a*). Then, the *Fad2* gene was introduced into the pCMV–Fad3 vector at the *Sal*I and *Not*I restriction sites, and this was named pCMV–Fad2–Fad3 (figure 3*a*). The primer information was presented in electronic supplementary material, table S5.

## 2.3. Generation of transgenic mice

The pronuclear microinjection was carried out as described previously [28,29]. Briefly, 6–8-week-old BDF1 female mice were superovulated through intraperitoneal (i.p.) injection of pregnant mare serum gonadotropin (10 IU; Sansheng, Ningbo, China) and human chorionic gonadotropin (hCG, 10 IU; Sansheng), 48 h apart. After the hCG i.p. injection, the female mice mated with male BDF1 mice in a 1 : 1 ratio in single cages overnight. To obtain pseudo-pregnant surrogate mice, two CD1/KM females were placed together with one vasectomized male overnight. The next morning, successful mating was confirmed by the presence of vaginal plugs. The plugged female mice were sacrificed through cervical dislocation, and zygotes were collected from the oviducts using a stereoscopic microscope (Nikon, Tokyo, Japan). To prepare DNA for microinjection, pCMV–Fad3 and pCMV–Fad2–Fad3 expression vectors were linearized and purified using the Plasmid Plus Kit (Qiagen) in accordance with the manufacturer's protocol. After purification, the DNA was diluted to $3 \, \text{ng} \, \mu\text{l}^{-1}$ in modified TE buffer (10 mM Tris–HCl, pH 7.5, 0.1 mM EDTA). The injection needle was filled with 1 µl of the DNA suspension. The linearized vector was injected into the well-recognized pronuclei. After injection, zygotes were maintained at room temperature for 30 min and then moved into the incubator. The injected zygotes were transferred into pseudo-pregnant female mice (approx. 30 zygotes per mouse) after a 2 h recovery culture in KSOM-AA medium. After 19–21 days, the mice pups were delivered naturally. For founder identification, the tail tips (approx. 1 cm) were subjected to standard DNA extraction procedures. The amplified DNA fragments were subjected to TA cloning and sequencing. The founder (F0) mice were crossed with their littermates to produce F1 mice. The primer information was presented in electronic supplementary material, table S5.

## 2.4. Determination of gene copy numbers in transgenic mice

Transgene copy number was detected by a standard protocol as previously described [30]. Briefly, transgene copy number was estimated by real-time quantitative polymerase chain reaction (qPCR). The qPCR was performed using a SYBR Premix Ex Taq (Takara) and signals were detected with Applied Biosystems 7500 real-time PCR system (Thermo, Waltham, MA, USA). The conditions used in PCR reactions were as follows: initial denaturation at 95°C for 30 s; 40 cycles of 95°C for 5 s, 60°C for 34 s. The exogenous gene copies by calculating the Fad3 and Gapdh copy number ratios. The primer information was presented in electronic supplementary material, table S5.

## 2.5. RT-PCR and qPCR analysis

Total RNA was isolated using TRIzol reagent (Thermo) and was immediately reverse-transcribed using a Prime Script RT reagent kit (Takara). The reverse transcription PCR (RT-PCR) was amplified using Ex Taq (Takara). The qPCR was performed using an SYBR Premix Ex Taq (Takara) and signals were detected with Applied Biosystems 7500 real-time PCR System (Thermo). Relative mRNA expression was calculated using the $2^{(-\Delta\Delta Ct)}$ method. The primer information is presented in electronic supplementary material, table S5.

## 2.6. PUFA analysis

The PUFA analysis was performed as previously described [17]. Briefly, wild-type and transgenic mice were assessed through gas chromatography, as previously reported [31]. Fresh mouse tissues were homogenized through grinding in liquid nitrogen, and an aliquot of the tissue homogenate in a glass methylation tube was mixed with 2 ml of chloroform-methanol (2 : 1). The samples were dried using nitrogen at 60°C. The precipitate was dissolved in 2 ml of pure hexane for chromatographic assessment, and then methyl esterification was added in a 400 µl saturated KOH methanol solution, mixed well, vortexed for 5 min and centrifuged for 10 min at $2000 \, \text{rpm} \, \text{min}^{-1}$. The gas chromatography coupled to mass spectrometry (GC-MS) analysis was performed using a GCMS-QP2010 Ultra (Shimadzu, Kyoto, Japan), and the conditions were as follows: hp-88 column, helium gas carrier, a constant linear velocity of $20.0 \, \text{cm} \, \text{s}^{-1}$, a segregation ratio of 20.0%, an injection volume of 1 µl, and a temperature programme of 60°C for 1 min, increased at $40°C \, \text{min}^{-1}$, and held at 140°C for 10 min, increased at $4°C \, \text{min}^{-1}$, and held at 240°C for 15 min to complete the run.

## 2.7. Statistical analyses

All experiments were repeated at least three times. One-way ANOVA was used to determine statistical significance following by Duncan's test to determine the statistical significance between the relative group. Statistical analysis was conducted using Statistical Analysis System software (SAS Institute, Cary, NC, USA). All data were expressed as mean ± s.d. Differences were considered to be significant when $p < 0.05$.

# 3. Results

## 3.1. Generation of Fad3 transgenic mice

We microinjected 75 fertilized zygotes with linearized Fad3 transgenic expression cassettes (figure 1a). Two recipients transferred with a total of 60 embryos became pregnant and gave birth to 16 offspring (26.7%). The PCR genotyping analyses showed that 3 of the 16 mice were positive (18.8%; 1 male and 2 females; figure 1b,c). As estimating the transgene copy number is a significant step in transgenic animal research, we next investigated the Fad3 gene in the transgenic mice. The real-time qPCR, based on serial dilution curves, was used to determine the copy number of exogenous Fad3 in transgenic mice, and the glyceraldehyde-3-phosphate dehydrogenase gene (Gapdh) in mice was used as an endogenous reference gene. With a serial of dilutions, the standard curves of the threshold cycle (Ct) relative to the log of each initial template copy of Fad3 and Gapdh gene were obtained, and the correlation coefficients were 0.9996 and 0.9997, respectively (electronic supplementary material, figure S1b and table S1). The transgenic copy number was obtained by comparing the initial template copy of Fad3 with that of Gapdh. Among the three putative transgenic mice, one male had 13 copy numbers, and two females had

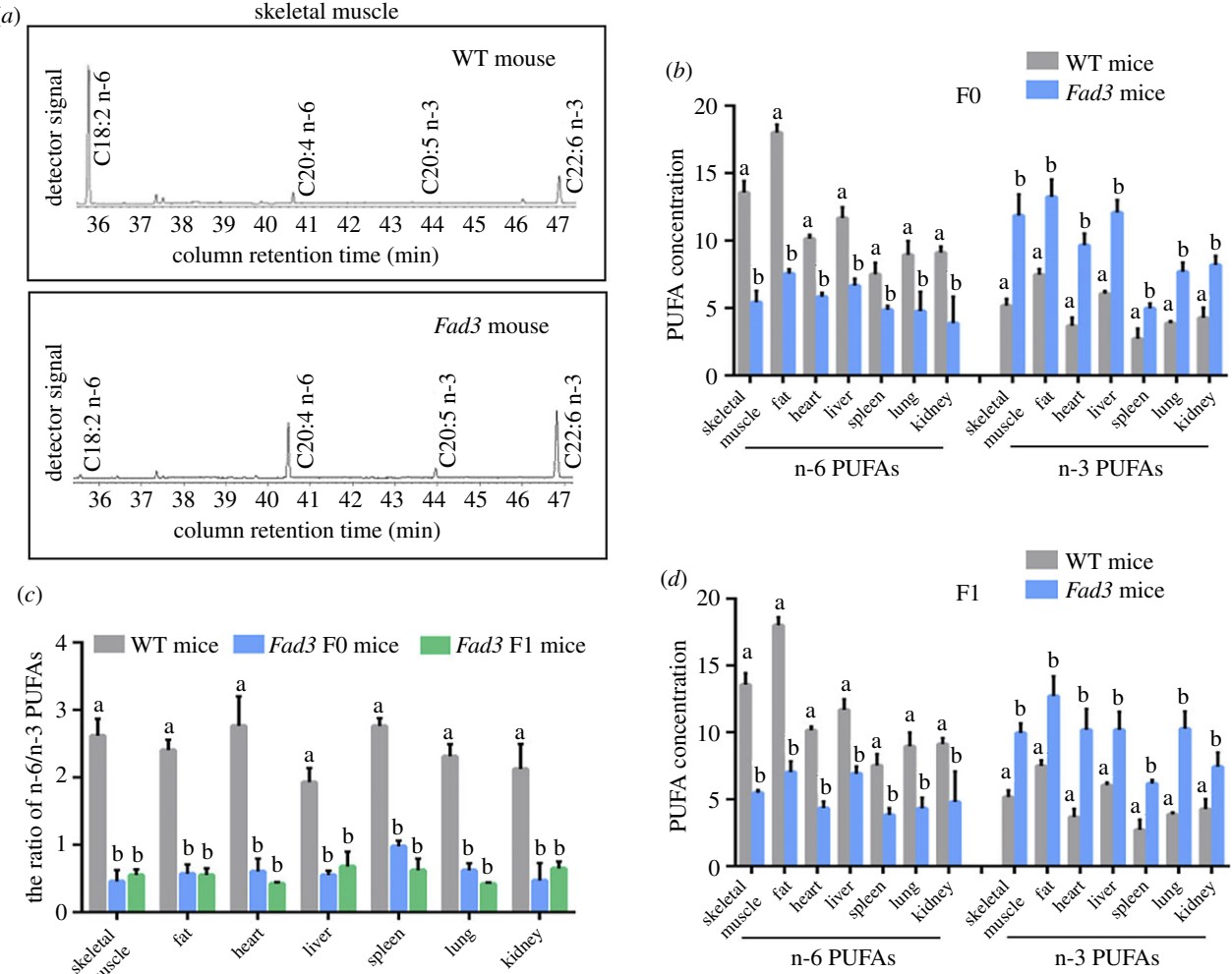

**Figure 2.** PUFA composition of *Fad3* single transgenic mice. (*a*) Partial gas chromatogram traces showed fatty acid profiles of total lipid extracted from muscle tissues of wild-type and *Fad3* single transgenic mice. (*b*) Comparison of the n-6 and n-3 PUFAs concentration between wild-type and *Fad3* single transgenic mice in skeletal muscle, fat, heart, liver, spleen, lung and kidney. (*c*) Ratio of n-6/n-3 PUFAs in wild-type and *Fad3* single transgenic mice. All the bars represent mean ± s.d., and different superscript letters (a–b) in each column represent statistically significant differences (*p* < 0.05), WT: wild-type.

14 and 5 copies, respectively (electronic supplementary material, table S2). In order to maintain the stability of the exogenous gene, the transgenic mice with the highest copies of *Fad3* were selected as F0 generation. At 6 weeks old, the founder F0 mice were crossed to the littermates to produce F1 mice (three males and five females). To assess the potential expression of the *Fad3* gene *in vivo*, we extracted the total RNA from F0 and F1 transgenic mice and analysed the mRNA by RT-PCR. As expected, the *Fad3* mRNA could be detected in the skeletal muscle, fat, heart, liver, spleen, lung and kidney of the F0 and F1 transgenic mice (figure 1*d*).

### 3.2. *Fad3* can convert n-6 PUFAs to n-3 PUFAs

To determine whether the plant-derived *Fad3* gene can convert n-6 to n-3 PUFAs in transgenic mice, the major tissues from transgenic mice and wild-type mice were assessed for n-6 PUFA (LA, γ-LA and AA) and n-3 PUFA (ALA, EPA and DHA) levels. Gas chromatographic analysis showed that F0 *Fad3* transgenic mice contained higher amounts of n-3 PUFAs and lower amounts of n-6 PUFAs compared with the wild-type mice (figure 2*a,b*; electronic supplementary material, table S3). In addition, the major tissues (skeletal muscle, fat, heart, liver, spleen, lung and kidney)

from F1 transgenic mice were also collected and analysed for n-3 and n-6 PUFAs. There was at least 2.84-fold reduction of the n-6/n-3 ratio in F1 transgenic mice compared with wild-type mice (skeletal muscle: from 2.62 to 0.55, *p* < 0.05; fat: from 2.41 to 0.55, *p* < 0.05; heart: from 2.77 to 0.43, *p* < 0.05; liver: from 1.93 to 0.68, *p* < 0.05; spleen: from 2.76 to 0.62, *p* < 0.05; lung: from 2.31 to 0.42, *p* < 0.05 and kidney: from 2.13 to 0.65, *p* < 0.05; figure 2*c*; electronic supplementary material, table S3). Moreover, the ratio of n-6/n-3 PUFAs in the F1 transgenic mice was similar to the ratio in F0 (skeletal muscle: from 2.62 to 0.46, *p* < 0.05; fat: from 2.41 to 0.57, *p* < 0.05; heart: from 2.77 to 0.61, *p* < 0.05; liver: from 1.93 to 0.55, *p* < 0.05; spleen: from 2.76 to 0.98, *p* < 0.05; lung: from 2.31 to 0.62, *p* < 0.05 and kidney: from 2.13 to 0.47, *p* < 0.05; figure 2*c*; electronic supplementary material, table S3). These results indicate that the plant *Fad3* gene can be functionally expressed in the major tissues of F0 and F1 transgenic mice, and *Fad3* plays an active role in the conversion of n-6 into n-3 PUFAs.

### 3.3. Generation of *Fad2–Fad3* double transgenic mice

Because *Fad3* single transgenic mice lack the Fad2 enzyme and cannot *de novo* synthesize n-6 PUFAs, they must take it from plants or seafood in their diets (i.e. the synthesis of

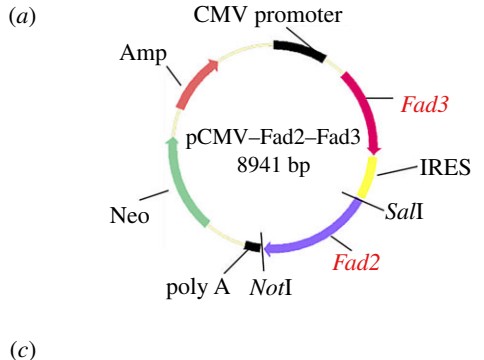

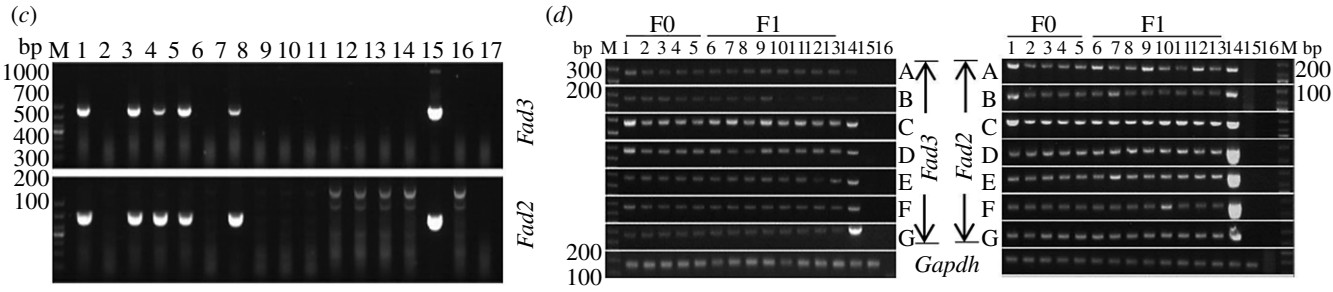

**Figure 3.** Generation of transgenic mice that express both *Fad2* and *Fad3* genes. (*a*) Schematic representation of pCMV–Fad2–Fad3 plasmid. (*b*) Summary of transgenic mice that carried *Fad2–Fad3* genes. (*c*) Identification of *Fad2–Fad3* transgenic mice by PCR (detection of *Fad2* and *Fad3* gene 573 and 865 bp; M: Marker DL1000; lanes 1–14: offspring mice; lane 15: positive control using wild-type mixed with pCMV–Fad3 plasmid as template; lane 16: negative control using wild-type genomic DNA as template; lane 19: blank control using $H_2O$ as template). (*d*) Validation of *Fad3* transgene expression in seven major tissues by RT-PCR (the PCR product for *Fad2*, *Fad3* and *Gapdh* cDNA is 191 bp, 235 bp and 193 bp, respectively; M: Marker DL1000; lanes 1–13: transgenic mice; lane 14: positive control using pCMV–Fad2–Fad3 plasmid as template; lane 15: negative control using wild-type cDNA as template; lane 16: blank control using $H_2O$ as template; A: skeletal muscle; B: fat; C: heart; D: liver; E: spleen; F: lung; G: kidney).

n-3 PUFAs is limited by n-6 PUFAs). We therefore microinjected a total number of 78 fertilized embryos, collected from superovulated female mice, with linearized pCMV–Fad2–Fad3 expression cassettes (figure 3*a*). Two recipients receiving a total number of 60 embryos became pregnant and gave birth to 14 (14/60, 23.3%) offspring. Genotyping analysis of the genomic DNA from each offspring indicated that five (5/14, 35.7%) of the puppy mice were *Fad2–Fad3* double transgenic (1 male and 4 females; figure 3*b,c*). Consistent with the strategy of single *Fad3* single transgenic mice breeding, the *Fad2–Fad3* mice were crossed with the positive littermate mice to produce F1 mice. To assess the expression potential of the plant *Fad2–Fad3* transgene *in vivo*, we extracted the total RNA from major tissues, including skeletal muscle, fat, heart, liver, spleen, lung and kidney and analysed the mRNA by RT-PCR with specific primers for *Fad2–Fad3* and housekeeping gene *Gapdh*. RT-PCR analysis indicated that the plant *Fad2–Fad3* genes were expressed in F0 and F1 transgenic mice, while no PCR signal was detected in the non-transgenic littermates (figure 3*d*).

### 3.4. *Fad2–Fad3* double transgenic mice can simultaneously produce both n-6 and n-3 PUFAs

To determine whether or not the double transgenic mice expressing the Fad2 enzyme catalysed the synthesis of n-6 PUFAs, the F0 double transgenic mice and non-transgenic littermates were used for PUFA analysis. The concentrations of total n-6 PUFAs (LA, γ-LA and AA) in major tissues of the *Fad2–Fad3* mice were at least 1.27-fold higher than in the wild-type mice (figure 4*a,b*). Among them, LA, γ-LA and AA showed at least 1.24-fold, 1.71-fold and 1.43-fold increase

in major tissues, respectively (figure 4*c*; electronic supplementary material, table S4). On the other hand, the concentration of total n-3 PUFAs (ALA, EPA and DHA) in the *Fad2–Fad3* transgenic was increased at least 2.74-fold (skeletal muscle: from ($5.18 \pm 0.49$) to ($20.77 \pm 1.38$), $p < 0.05$; fat: from ($7.48 \pm 0.41$) to ($20.47 \pm 1.37$), $p < 0.05$; heart: from ($3.67 \pm 0.62$) to ($16.63 \pm 0.24$), $p < 0.05$; liver: from ($6.05 \pm 0.21$) to ($18.64 \pm 0.89$), $p < 0.05$; spleen: from ($2.72 \pm 0.76$) to ($9.54 \pm 0.90$), $p < 0.05$; lung: from ($3.87 \pm 0.16$) to ($14.09 \pm 1.16$), $p < 0.05$ and kidney: from ($4.28 \pm 0.75$) to ($15.55 \pm 1.30$), $p < 0.05$; figure 4*b*; electronic supplementary material, table S4). For the detailed analysis of n-3 PUFAs, the concentrations of ALA, EPA and DHA were significantly increased in *Fad2–Fad3* mouse skeletal muscle (ALA: 5.76-fold; EPA: 5.05-fold; DHA: 3.83-fold), fat (ALA: 1.91-fold; EPA: 4.5-fold; DHA: 2.74-fold), heart (ALA: 1.72-fold; EPA: 4.15-fold; DHA: 4.83-fold), liver (ALA: 4.87-fold; EPA: 5.2-fold; DHA: 2.89-fold), spleen (ALA: 2-fold; EPA: 2.05-fold; DHA: 3.71-fold), lung (ALA: 2.91-fold; EPA: 5.22-fold; DHA: 3.64-fold) and kidney (ALA: 3.54-fold; EPA: 2.74-fold; DHA: 3.71-fold) compared with the wild-type mouse (figure 4*d*; electronic supplementary material, table S4).

To assess the function of the Fad2–Fad3 enzymes in the F0 and F1 transgenic mouse, the PUFA content in the major tissues was compared, including skeletal muscle, heart, liver, spleen, lung, kidney and fat. Figure 4*b* and electronic supplementary material, table S4 display the PUFA profile of seven tissues from F0 and F1 of the *Fad2–Fad3* mouse. It shows that all the n-6 and n-3 PUFAs were higher when compared with wild-type mice, indicating that the *Fad2–Fad3* double transgenic mice had efficiently converted the monounsaturated fatty acids into n-6 and n-3 PUFAs in their bodies.

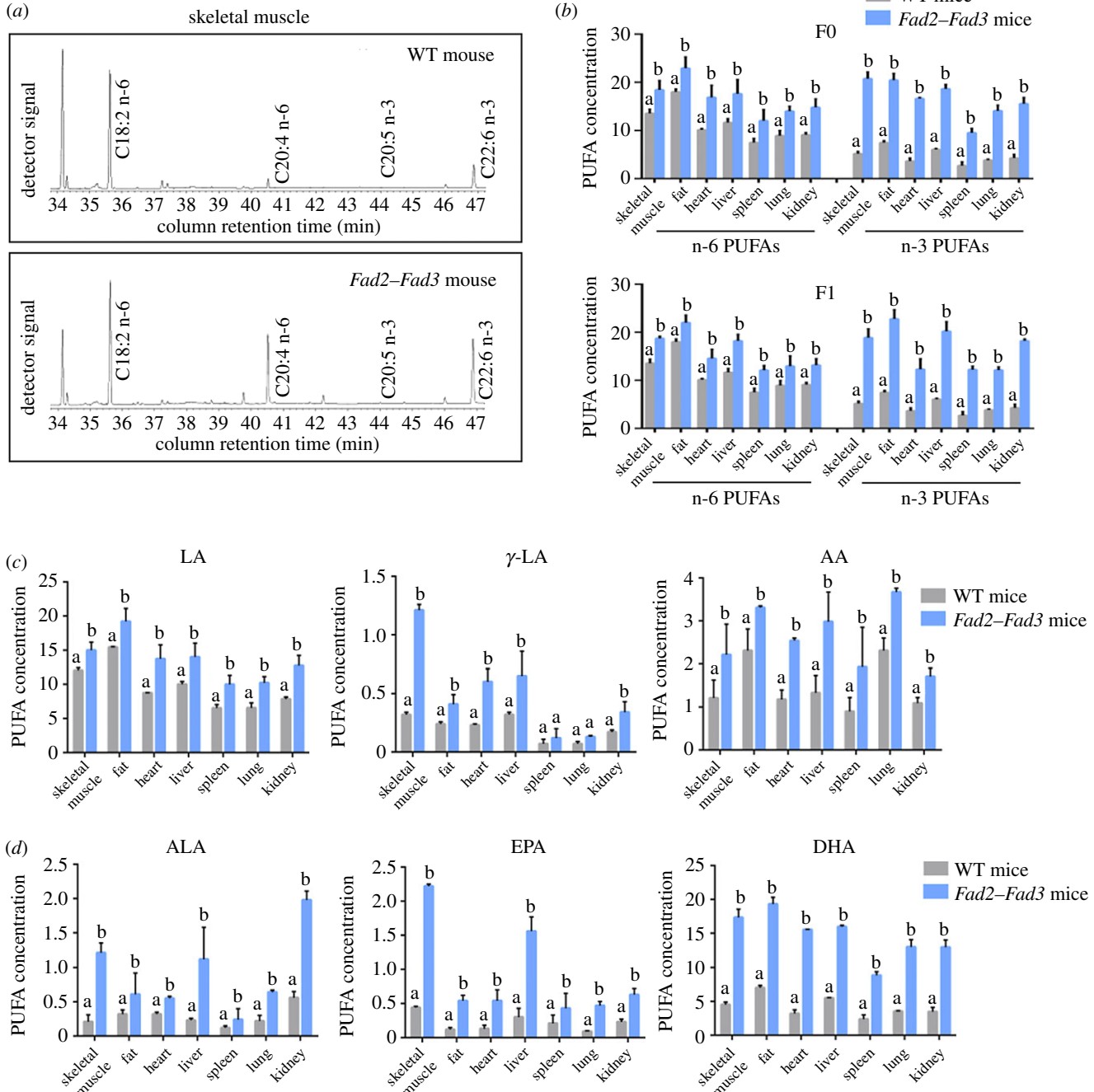

**Figure 4.** PUFA composition of *Fad2–Fad3* double transgenic mice. (*a*) Partial gas chromatogram traces showing fatty acid profiles of total lipid extracted from muscle tissues of wild-type (WT) and *Fad2–Fad3* double transgenic mice. (*b*) Comparison of the PUFA concentration between wild-type and *Fad2–Fad3* double transgenic mice in skeletal muscle, fat, heart, liver, spleen, lung and kidney. (*c*) The concentration analysis of n-6 PUFAs (LA, γ-LA and AA) of wild-type and *Fad2–Fad3* double transgenic mice. (*d*) The n-3 PUFAs (ALA, EPA and DHA) concentration in wild-type and *Fad2–Fad3* double transgenic mice. All the bars represent mean ± s.d., and different superscript letters (a–b) in each column represent statistically significant differences (*p* < 0.05).

## 3.5. *Fad2–Fad3* double transgenic mice established their own PUFA biosynthetic pathways

To verify the biological functions of *Fad2* and *Fad3* in the *Fad2–Fad3* transgenic mice, we examined the expression level of fatty acid biosynthesis-related genes, including fatty acid synthase genes, fatty acid-binding protein and fatty acid oxidation genes. Compared with *Fad3* single transgenic and the wild-type mice, the fatty acid synthesis genes including *Fasn*, *Scd1* and *Acc* were significantly decreased in the *Fad2–Fad3* double transgenic mice, as detected by qPCR (figure 5*a*). We also compared the fatty acid-binding protein (*Fabp4*) and

fatty acid oxidation-related genes (*Lipe*, *Lpl*, *Ppar-γ*, *Lcad*) expression between different transgenic mice via qPCR. The qPCR results showed that the expression of fatty acid-binding protein and fatty acid oxidation-related genes were increased in *Fad3* single and *Fad2–Fad3* double transgenic mice compared with the wild-type mice (figure 5*a*).

To further consolidate the qPCR results, we compared the concentrations of total n-6 and n-3 PUFAs between different transgenic mice. The seven major tissues from *Fad2–Fad3* double transgenic mice were collected and analysed for n-6/n-3 PUFA ratio. Notably, F0 and F1 double transgenic mice showed a substantially lower n-6/n-3 ratio in all tissues examined compared with wild-type mice (skeletal muscle:

royalsocietypublishing.org/journal/rsob  Open Biol. 9: 190140

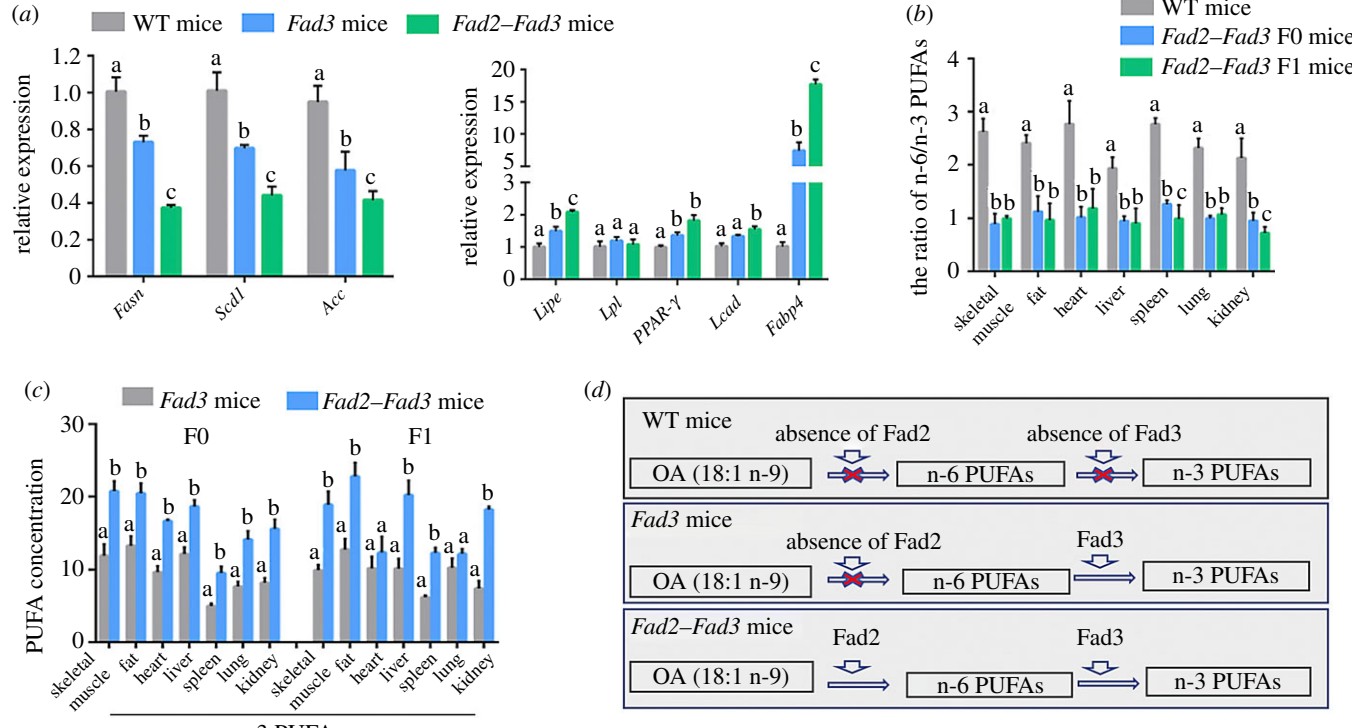

**Figure 5.** *Fad2–Fad3* double transgenic mice have their own PUFA biosynthetic pathways. (*a*) RT-qPCR analysis for fatty acid-related genes in wild-type and *Fad2–Fad3* double transgenic mice. Data shown are mean expression values relative to *Gapdh*; the value in wild-type control was set as 1; different superscript letters (a–c) in each column represent statistically significant differences, $p < 0.05$; the bars represent mean ± s.d. (*b*) The ratio of n-6/n-3 PUFAs in wild-type and *Fad2–Fad3* double transgenic mice. Different superscript letters (a–c) in each column represent statistically significant differences, $p < 0.05$; the bars represent mean ± s.d. (*c*) The different n-3 PUFA content of *Fad3* single and *Fad2–Fad3* double transgenic mice. Different superscript letters (a–b) in each column represent statistically significant differences, $p < 0.05$; the bars represent mean ± s.d. (*d*) Illustration of the mechanisms of Fad enzymes on the PUFA biosynthetic pathway. The Fad2 and Fad3 enzymes are functional in converting n-9 to n-6 PUFAs and converting n-6 to n-3 PUFAs, respectively. Due to a lack of Fad2 and Fad3 enzymes, wild-type mice are unable to synthesis n-6 and n-3 PUFAs. Here, we report the *Fad2–Fad3* double transgenic mice that produce high levels of both n-6 and n-3 PUFAs in their tissues and organs.

0.99, 0.89 versus 2.61; fat: 1.12, 0.97 versus 2.41; heart: 1.18, 1.02 versus 2.77; liver: 0.90, 0.95 versus 1.93; spleen: 0.98, 1.26 versus 2.76; lung: 1.07, 1.00 versus 2.31; kidney: 0.72, 0.95 versus 2.13; figure 5*b*; electronic supplementary material, table S4). Compared with *Fad3* single transgenic mice, the concentrations of total n-3 PUFAs (ALA, EPA and DHA) in major tissues of the *Fad2–Fad3* mice were increased (figure 5*c*). These results indicated that plant *Fad2* and *Fad3* genes are functionally expressed in transgenic mice. The *Fad2–Fad3* double transgenic mice have successfully established their own PUFA biosynthetic pathways, in which Fad2 enzyme *de novo* synthesizes n-6 PUFAs, and the Fad3 enzyme further converts n-6 into n-3 PUFAs (figure 5*d*).

## 4. Discussion

Except for *C. elegans*, animals (including humans) have not been reported to possess desaturase genes, which can change the ratio of n-6/n-3 fatty acids [1,4]. Mammals must obtain n-6 PUFAs and n-3 PUFAs from their daily diet. In order to enable humans to take in more health beneficial PUFAs from the dietary food, an increasing number of researchers began to produce transgenic livestock that carries fatty acid desaturase genes.

In recent years, we have produced *Fat1* transgenic cows and sheep [18,22,23]. An analysis of fatty acids demonstrated that the *Fat1* transgenic animals produced high levels of n-3

PUFAs and a significantly reduced n-6/n-3 PUFA ratio in their tissues and milk [21,23,32–34]. Lai *et al.* also reported that EPA (20 : 5, n-3) and DHA (22 : 6, n-3) levels in *Fat1* transgenic pigs increased by 15-fold and 4-fold, respectively, and the n-6/n-3 PUFAs ratio decreased by 80.2% [20,35]. Therefore, the n-3 PUFAs desaturase Fat1 can be functionally expressed in animals, or at least in cows, sheep and pigs. The n-3 PUFAs contents in *Fat1* transgenic animals were significantly improved, which may make the n-3 PUFAs rich animal source food for human nutrition. However, transgenic animals prepared using *C. elegans*-derived *Fat1* genes may create problems in terms of food safety [13].

Previous studies of *Fat1* transgenic animals have mainly focused on accelerating PUFA transition from n-6 to n-3 PUFAs. Although *Fat1* transgenic animals are rich in n-3 PUFAs, these animals lack n-6 PUFAs [21,23,32–34]. Deficiencies in n-6 PUFAs may cause reproductive failure, skin lesions, fatty liver, reduced growth and polydipsia [25]. To solve these problems, we used plant *Fad2 gene*, which can promote *de novo* biosynthesis of n-6 PUFAs [17,27]. Although the *Fat1* gene has been widely used in transgenic animals, related research and application of the *Fad* gene have not been widely reported.

In the current study, we successfully produced transgenic mice that carried single *Fad3* or double *Fad2–Fad3* genes. Our results indicate that the plant-derived *Fad* transgenic system can endogenously synthesize PUFAs in a transgenic animal. In the *Fad3* single transgenic mice, we found that

the concentration of the n-6 PUFAs was decreased and that of the n-3 PUFAs was increased compared with wild-type mice. By contrast, in *Fad2–Fad3* double transgenic mice, the concentration of both n-6 and n-3 PUFAs was significantly increased compared with *Fad3* single transgenic and wild-type mice. Thus, the *Fad2–Fad3* double transgenic mice have their own PUFA biosynthetic pathways. These *Fad* transgenic mice provide an animal model with which to study the mechanism of fatty acid biosynthesis. In addition, the generation of *Fad2–Fad3* double transgenic livestock that produces PUFAs may be an economical and safe way to produce PUFA-rich food.

Ethics. All animal experiments were approved by the Animal Care and Use Committee of Inner Mongolia University and were performed in accordance with the Animal Research Institute Committee guidelines.

Data accessibility. This article has no additional data.

Authors' contributions. G.L. conceived of and designed the study. L.S., L.Y., J.W., X.L., L.B. and A.D. performed the experiments and analysed the data. G.L. and L.Y. supervised the project. L.S., L.Y. and G.L. wrote the manuscript. All authors read and approved the final manuscript.

Competing interests. All authors declare no competing interests.

Funding. This study was supported by the Genetically Modified Organisms Breeding Major Projects (grant no. 2016ZX08007-002), the opening project of State Key Laboratory of R2BGL (to L.Y.), the Inner Mongolia Autonomous Region Basic Research Project (to G.L.) and Inner Mongolia Science and Technology Innovation Guide Project (KCBJ2018002).

Acknowledgements. We are grateful to our colleagues in the laboratory for their assistance with the experiments and in the preparation of this manuscript.

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
