## [Reviewer comments · Open Biology]

Review History

RSOB-19-0140.R0 (Original submission)

Review form: Reviewer 1

Recommendation

Major revision is needed (please make suggestions in comments)

Do you have any ethical concerns with this paper?

No

Comments to the Author

Comments: "Generation of *Fad2* and *Fad3* transgenic mice that produce n-6 and n-3 polyunsaturated fatty acids" by Song et al. generated transgenic mice that expressed plant gene *Fad2* and *Fad3*, and thus could produce n-6 and n-3 PUFAs. There are my major concerns mentioned below.

1. Line 94: Readers need to know the plant samples you used to amplified *Fad2* and *Fad3*, and the method to extarct the plant RNA. The authors should add these details.
2. Line 97, Line 109, Line 110, Line 116: Thermo, Promega, TAKARA are not Chinese

brands/companies.

3. Line 115: "all kinds of tissues" is overstated. The authors only investigated liver and muscle from transgenic and wild-type mice.

4. Line 145, Line 146, Line 176: "various tissues" and "transgenic tissues" are not precise. Figure 1c and 2c only show the RT-PCR results of liver.

5. Line 152, Line 394, Line 395 and Figure 1d-1e: The authors talked about PUFA analysis of liver samples in F0 transgenic mice in the text, but the legend of Figure 1d showed it was muscle samples, and it was unclear of which samples used in Figure 1e. Same mistakes were found in Figure 2d-2e.

The authors need to display the PUFA analysis results of both samples (i.e. liver and muscle) in Figure 1 and 2, and mark the sample name in the figures.

6. Figure 1e, 2e, 3d and 3e: The results of PUFA analysis in Figure 3d and 3e have error bars, but not shown in Figure 1e and 2e. The authors should not omit this.

7. What is the difference in the results of PUFA analysis between F0 and F1 transgenic mice? The authors need to address this critical point.

8. There are a few grammar and word mistakes that should be corrected. We recommend that the manuscript is edited by a native speaker, to improve its flow and clarity.

Decision letter (RSOB-19-0140.R0)

05-Aug-2019

Dear Professor Li,

We are writing to inform you that the Editor has reached a decision on your manuscript RSOB-19-0140 entitled "Generation of Fad2 and Fad3 transgenic mice that produce n-6 and n-3 polyunsaturated fatty acids", submitted to Open Biology.

As you will see from the reviewer's comments below, there are a number of criticisms that prevent us from accepting your manuscript at this stage. The reviewer suggests, however, that a revised version could be acceptable, if you are able to address their concerns. If you think that you can deal satisfactorily with the suggestions, we would be pleased to consider a revised manuscript.

The revision will be re-reviewed, where possible, by the original referees. As such, please submit the revised version of your manuscript within six weeks. If you do not think you will be able to meet this date please let us know immediately.

When submitting your revised manuscript, please respond to the comments made by the referee(s) and upload a file "Response to Referees" in "Section 6 - File Upload". You can use this to document any changes you make to the original manuscript. In order to expedite the

processing of the revised manuscript, please be as specific as possible in your response to the referee(s).

Please see our detailed instructions for revision requirements
<https://royalsociety.org/journals/authors/author-guidelines/>

Sincerely,

The Open Biology Team

Reviewer Comments to Author(s):

Referee:

Comments to the Author(s)

Comments: "Generation of Fad2 and Fad3 transgenic mice that produce n-6 and n-3 polyunsaturated fatty acids" by Song et al. generated transgenic mice that expressed plant gene Fad2 and Fad3, and thus could produce n-6 and n-3 PUFAs. There are my major concerns mentioned below.

1. Line 94: Readers need to know the plant samples you used to amplified Fad2 and Fad3, and the method to extarct the plant RNA. The authors should add these details.
2. Line 97, Line 109, Line 110, Line 116: Thermo, Promega, TAKARA are not Chinese brands/companies.
3. Line 115: "all kinds of tissues" is overstated. The authors only investigated liver and muscle from transgenic and wild-type mice.
4. Line 145, Line 146, Line 176: "various tissues" and "transgenic tissues" are not precise. Figure 1c and 2c only show the RT-PCR results of liver.
5. Line 152, Line 394, Line 395 and Figure 1d-1e: The authors talked about PUFA analysis of liver samples in F0 transgenic mice in the text, but the legend of Figure 1d showed it was muscle samples, and it was unclear of which samples used in Figure 1e. Same mistakes were found in Figure 2d-2e.
The authors need to display the PUFA analysis results of both samples (i.e. liver and muscle) in Figure 1 and 2, and mark the sample name in the figures.
6. Figure 1e, 2e, 3d and 3e: The results of PUFA analysis in Figure 3d and 3e have error bars, but not shown in Figure 1e and 2e. The authors should not omit this.
7. What is the difference in the results of PUFA analysis between F0 and F1 transgenic mice? The authors need to address this critical point.
8. There are a few grammar and word mistakes that should be corrected. We recommend that the manuscript is edited by a native speaker, to improve its flow and clarity.

Author's Response to Decision Letter for (RSOB-19-0140.R0)

See Appendix A.

RSOB-19-0140.R1 (Revision)

Review form: Reviewer 1

Recommendation

Accept as is

Do you have any ethical concerns with this paper?

No

Comments to the Author

The authors have addressed the concerns raised by my review. So it can be accepted for publication.

Decision letter (RSOB-19-0140.R1)

20-Sep-2019

Dear Professor Li

We are pleased to inform you that your manuscript entitled "Generation of Fad2 and Fad3 transgenic mice that produce n-6 and n-3 polyunsaturated fatty acids" has been accepted by the Editor for publication in Open Biology.

If applicable, please find the referee comments below. No further changes are recommended.

Sincerely,

The Open Biology Team
mailto:openbiology@royalsociety.org

Reviewer(s)' Comments to Author:

Referee: 1

Comments to the Author(s)

The authors have addressed the concerns raised by my review. So it can be accepted for publication.

Appendix A

Dear Editor,

I, along with my coauthors, would like to re-submit the attached manuscript entitled “Generation of Fad2 and Fad3 transgenic mice that produce n-6 and n-3 polyunsaturated fatty acids” as a Research Article in the *Open Biology*. The manuscript ID is **RSOB-19-0140**.

The manuscript has been carefully rechecked and appropriate changes have been made in accordance with the reviewers’ suggestions. The responses to their comments have been prepared and attached herewith. The revised manuscript has been checked by a native English speaker to ensure the language quality of our manuscript. Furthermore, the paper has been edited by *Charlesworth Language Editing Services*, and the CERTIFICATE is shown below.

We thank you and the reviewers for your thoughtful suggestions and insights, which have enriched the manuscript and produced a more balanced and better account of the research. We hope that the revised manuscript is now suitable for publication in *Open Biology*.

Thank you for your consideration. I look forward to hearing from you.

Sincerely,

Guangpeng Li

Research Center for Mammalian Reproductive Biology and Biotechnology

Inner Mongolia University

Hohhot 010070, China

email: gpengli@imu.edu.cn; gpengli@hotmail.com

LANGUAGE-CERTIFICATE

EDITORIAL CERTIFICATE

This document certifies that the manuscript below was edited for correct English language usage, grammar, punctuation and spelling by qualified native English speaking editors at Charlesworth Author Services.

Paper Title:

Generation of Fad2 and Fad3 transgenic mice that produce n-6 and n-3 polyunsaturated fatty acids

Author:

Lei Yang

Date certificate issued:

August 23, 2019

cwaitors.com

POINT-BY-POINT RESPONSES TO THE REVIEWS

Manuscript ID: RSOB-19-0140

First of all, we thank both reviewers for taking the time to read and comment on our manuscript. We also thank the Editor for their clear guidance.

Response to Reviewer:

Comments: “Generation of Fad2 and Fad3 transgenic mice that produce n-6 and n-3 polyunsaturated fatty acids” by Song et al. generated transgenic mice that expressed plant gene Fad2 and Fad3, and thus could produce n-6 and n-3 PUFAs. There are my major concerns mentioned below.

Response:

We greatly appreciate the reviewer’s valuable comments regarding our manuscript. The manuscript has been completely rewritten and further edited by an English native speaker as recommended.

Comment 1:

Line 94: Readers need to know the plant samples you used to amplified Fad2 and Fad3, and the method to extarct the plant RNA. The authors should add these details.

Response 1:

We apologize for this mistake and thank the reviewer for bringing it to our attention. We have added a description of the Fad2 and Fad3 vector in the revised “Materials and Methods” section.

(revised manuscript page 4, line 85–87) “The *Fad2* and *Fad3* gene used in this study were PCR-amplified from the cDNA of *Spinacia oleracea* and *Linum usitatissimum* (also known as common-flax or linseed), respectively. Total RNA was extracted from *S. oleracea* and *L. usitatissimum* using the RNeasy Plant Mini Kit (Qiagen, Hilden, Germany).”

Comment 2:

Line 97, Line 109, Line 110, Line 116: Thermo, Promega, TAKARA are not Chinese brands/companies.

Response 2:

We apologise for this careless mistake. In the revised manuscript, companies mentioned in the “Materials and Methods” include the name of the company, and the relevant city, state/province, and country.

(revised manuscript page 4, line 87) “RNeasy Plant Mini Kit (Qiagen, Hilden, Germany)”

(revised manuscript page 5, line 90) “pMD19-T vector (Takara, Kusatsu, Japan)”

(revised manuscript page 5, line 105-106) “stereoscopic microscope (Nikon, Tokyo, Japan)”

(revised manuscript page 6, line 124-125) “Applied Biosystems 7500 real-time PCR System (Thermo, Waltham, MA, USA)”

Comment 3:

Line 115: “all kinds of tissues” is overstated. The authors only investigated liver and muscle from transgenic and wild-type mice.

Response 3:

We agree that the phrase “all kinds of tissues” overstated the original number of investigated tissue types. We apologize for the results improper description.

In the revised manuscript, we re-performed the gas-chromatographic assay in seven major tissues of the *Fad3* single transgenic mice (F0 & F1), *Fad2-Fad3* double transgenic mice (F0 & F1), and wild-type mice (non-transgenic littermates). The major tissues include skeletal muscle, fat, heart, liver, spleen, lung, and kidney.

The newly revised sentences are displayed in **Response 5**.

Comment 4:

Line 145, Line 146, Line 176: “various tissues” and “transgenic tissues” are not precise. Figure 1c and 2c only show the RT-PCR results of liver.

Response 4:

Thanks to carefully read our manuscript. We agree that these sentences were not precise. In accordance with your suggestion, we re-performed the RT-PCR assay in seven major tissues of the *Fad3* single transgenic mice (F0 & F1), *Fad2-Fad3* double transgenic mice (F0 & F1), and wild-type mice (non-transgenic littermates).

In the revised manuscript, the new sentence reads as follows:

(revised manuscript page 8-9, line 174-177) “To assess the potential expression of the *Fad3* gene *in vivo*, we extracted the total RNA from F0 and F1 transgenic mice, and analyzed the mRNA by reverse transcription PCR (RT-PCR). As expected, the *Fad3* mRNA could be detected in the skeletal muscle, fat, heart, liver, spleen, lung, and kidney of the F0 and F1 transgenic mice (response Figure 1A, below; revised manuscript figure 1d)”.

(revised manuscript page 10, line 207-212) “To assess the expression potential of the plant *Fad2-Fad3* transgene *in vivo*, we extracted the total RNA from major tissues, including skeletal muscle, fat, heart, liver, spleen, lung, and kidney, and analyzed the mRNA by RT-PCR with specific primers for *Fad2-Fad3* and housekeeping gene *Gapdh*. RT-PCR analysis indicated that the plant *Fad2-Fad3* genes were expressed in F0 and F1 transgenic mice, while no PCR signal was detected in the non-transgenic littermates (response Figure 1B, below; revised manuscript figure 3d).

Figure 1 Response to Reviewer

- (A) Validation of *Fad3* transgene expression in seven major tissues by RT-PCR (the PCR product for *Fad3* and *Gapdh* cDNA is 235 bp and 193 bp, respectively; M: Marker DL1000; lanes 1 to 11: transgenic mice; lane 12: positive control using wild-type cDNA mixed with pCMV-*Fad3* plasmid as template; lane 13: negative control using wild-type cDNA as template; lane 14: blank control using H₂O as template; A: skeletal muscle; B: fat; C: heart; D: liver; E: spleen; F: lung; G: kidney).
- (B) Validation of *Fad3* transgene expression in seven major tissues by RT-PCR (the PCR product for *Fad2*, *Fad3*, and *Gapdh* cDNA is 191 bp, 235 bp, and 193 bp, respectively; M: Marker DL1000; lanes 1 to 13: transgenic mice; lane 14: positive control using pCMV-*Fad2*-*Fad3* plasmid as template; lane 15: negative control using wild-type cDNA as template; lane 16: blank control using H₂O as template; A: skeletal muscle; B: fat; C: heart; D: liver; E: spleen; F: lung; G: kidney).

Comment 5:

Line 152, Line 394, Line 395 and Figure 1d-1e: The authors talked about PUFA analysis of liver samples in F0 transgenic mice in the text, but the legend of Figure 1d showed it was muscle samples, and it was unclear of which samples used in Figure 1e. Same mistakes were found in Figure 2d-2e.

The authors need to display the PUFA analysis results of both samples (i.e. liver and muscle) in Figure 1 and 2, and mark the sample name in the figures.

We appreciate your insightful comment.

For the statistical results, we re-calculated all the experimental data, and modified the figure legends to improve clarity by adding information regarding the experimental approach.

For the PUFA analysis, we re-performed the gas-chromatographic assay in seven major tissues of the *Fad3* single transgenic mice (F0 & F1), *Fad2-Fad3* double transgenic mice (F0 & F1), and wild-type mice (non-transgenic littermates). We also added the source data in the “Supplementary Material” section (response Figure 2B-C, below; revised manuscript electronic supplementary material, table S3), (response Figure 2E-F, below; revised manuscript electronic supplementary material, table S4).

In the revised manuscript, the new sentence reads as follows:

1. (revised manuscript page 9, line 180-196) “To determine whether the plant derived *Fad3* gene can convert n-6 to n-3 PUFAs in transgenic mice, the major tissues from transgenic mice and wild-type mice were assessed for n-6 PUFA (LA, γ -LA and AA) and n-3 PUFA (ALA, EPA and DHA) levels. Gas chromatographic analysis showed that F0 *Fad3* transgenic mice contained higher amounts of n-3 PUFAs and lower amounts of n-6 PUFAs compared with the wild-type mice (response Figure 2A-B, below; revised manuscript figure 2a-b). In addition, the major tissues (skeletal muscle, fat, heart, liver, spleen, lung, and kidney) from F1 transgenic mice were also collected and analyzed for n-3 and n-6 PUFAs. There was at least 2.84-fold reduction of the n-6/n-3 ratio in F1 transgenic mice compared with wild-type mice (skeletal

muscle: from 2.62 to 0.55, $P < 0.05$; fat: from 2.41 to 0.55, $P < 0.05$; heart: from 2.77 to 0.43, $P < 0.05$; liver: from 1.93 to 0.68, $P < 0.05$; spleen: from 2.76 to 0.62, $P < 0.05$; lung: from 2.31 to 0.42, $P < 0.05$; and kidney: from 2.13 to 0.65, $P < 0.05$; (response Figure 2C, below; revised manuscript figure 2c).”

2. (revised manuscript page 10-11, line 215-232) “To determine whether or not the double transgenic mice expressing the *Fad2* enzyme catalyzed the synthesis of n-6 PUFAs, the F0 double transgenic mice and non-transgenic littermates were used for PUFA analysis. The concentrations of total n-6 PUFAs (LA, γ -LA and AA) in major tissues of the *Fad2-Fad3* mice were at least 1.27-fold higher than in the wild-type mice (response Figure 2D-E, below; revised manuscript figure 4a-b). Among them, LA, γ -LA and AA showed at least 1.24-fold, 1.71-fold and 1.43-fold increase in major tissues, respectively (response Figure 2F, below; revised manuscript figure 4c). On the other hand, the concentration of total n-3 PUFAs (ALA, EPA and DHA) in the *Fad2-Fad3* transgenic was increased at least 2.74-fold (skeletal muscle: from (5.18 ± 0.49) to (20.77 ± 1.38) , $P < 0.05$; fat: from (7.48 ± 0.41) to (20.47 ± 1.37) , $P < 0.05$; heart: from (3.67 ± 0.62) to (16.63 ± 0.24) , $P < 0.05$; liver: from (6.05 ± 0.21) to (18.64 ± 0.89) , $P < 0.05$; spleen: from (2.72 ± 0.76) to (9.54 ± 0.90) , $P < 0.05$; lung: from (3.87 ± 0.16) to (14.09 ± 1.16) , $P < 0.05$; and kidney: from (4.28 ± 0.75) to (15.55 ± 1.30) , $P < 0.05$; response Figure 2G, below; revised manuscript figure 4d).”

A

Skeletal muscle

B

C

D

E

F

Figure2 Response to Reviewer

- (A) Partial gas chromatogram traces showed fatty acid profiles of total lipid extracted from mouse muscle tissues of wild-type and *Fad3* single transgenic mice.
- (B) Comparison of the n-6 and n-3 PUFAs concentration between wild-type and *Fad3* single transgenic mice in skeletal muscle, fat, heart, liver, spleen, lung, and kidney.
- (C) The ratio of n-6/n-3 PUFAs in wild-type and *Fad3* single transgenic mice.
- (D) Partial gas chromatogram traces showing fatty acid profiles of total lipid extracted from mouse muscle tissues of wild-type and *Fad2-Fad3* double transgenic mice.
- (E) Comparison of the PUFA concentration between wild-type and *Fad2-Fad3* double transgenic mice in skeletal muscle, fat, heart, liver, spleen, lung, and kidney.
- (F) The concentration analysis of n-6 PUFAs (LA, γ -LA, and AA) of wild-type and *Fad2-Fad3* double transgenic mice.
- (G) The n-3 PUFAs (ALA, EPA, and DHA) concentration in wild-type and *Fad2-Fad3* double transgenic mice.
- All the bars represent mean \pm S.D., and different superscript letters (a-b) in each column represent statistical significant differences ($P < 0.05$), WT: wild-type.

Table S3. PUFA compositions of seven major tissues in *Fad3* F0/F1 transgenic mice and wild-type mice.

Skeletal muscle			Fat			Heart			Liver			Spleen			Lung			Kidney		
WT (n=8)	Fad3 F0(n=3)	Fad3 F1(n=8)	WT (n=8)	Fad3 F0(n=3)	Fad3 F1(n=8)	WT (n=8)	Fad3 F0(n=3)	Fad3 F1(n=8)	WT (n=8)	Fad3 F0(n=3)	Fad3 F1(n=8)	WT (n=8)	Fad3 F0(n=3)	Fad3 F1(n=8)	WT (n=8)	Fad3 F0(n=3)	Fad3 F1(n=8)	WT (n=8)	Fad3 F0(n=3)	Fad3 F1(n=8)
24.32±2.99 ^a	24.23±2.7 ^a	25.03±3.12 ^a	28.01±3.14 ^a	27.03±2.15 ^a	26.93±4.32 ^a	16.66±1.26 ^a	14.98±0.6 ^a	15.67±0.12 ^a	17.78±1.23 ^a	16.12±2.38 ^a	15.89±1.34 ^a	13.32±0.37 ^a	12.05±1.29 ^a	11.91±2.70 ^a	12.21±1.14 ^a	11.95±0.26 ^a	12.57±3.11 ^a	14.03±1.4 ^a	13.76±2.01 ^a	15.2
12.03±0.42 ^b	4.32±0.33 ^b	4.53±0.12 ^b	15.45±0.09 ^a	6.33±0.27 ^b	5.64±0.47 ^b	8.74±0.04 ^a	5.43±0.18 ^b	3.98±0.38 ^c	10.02±0.38 ^a	5.73±0.29 ^b	6.00±0.35 ^b	6.54±0.49 ^a	4.32±0.13 ^b	3.23±0.46 ^c	6.56±0.73 ^c	3.53±0.59 ^b	3.05±0.54 ^b	7.84±0.31 ^a	3.43±1.85 ^b	4.4
0.32±0.02 ^a	0.18±0.02 ^b	0.23±0.04 ^b	0.24±0.02 ^a	0.11±0.01 ^b	0.05±0.04 ^b	0.23±0.01 ^a	0.09±0.02 ^b	0.07±0.04 ^b	0.32±0.02 ^a	0.04±0.01 ^b	0.04±0.01 ^b	0.07±0.04 ^a	0.05±0.03 ^b	0.04±0.02 ^b	0.07±0.02 ^a	0.02±0.65 ^b	0.04±0.02 ^b	0.17±0.02 ^a	0.02±0.01 ^b	0.0
1.21±0.42 ^a	0.93±0.49 ^b	0.73±0.04 ^b	2.31±0.50 ^a	1.12±0.03 ^b	1.32±0.30 ^b	1.18±0.22 ^a	0.33±0.08 ^b	0.29±0.08 ^b	1.33±0.40 ^a	0.89±0.22 ^b	0.87±0.15 ^b	0.90±0.32 ^a	0.50±0.13 ^b	0.58±0.02 ^b	2.31±0.29 ^a	1.23±0.19 ^b	1.23±0.14 ^b	1.09±0.13 ^a	0.42±0.12 ^b	0.3
0.21±0.10 ^a	0.42±0.33 ^b	0.33±0.05 ^b	0.32±0.06 ^a	0.77±0.36 ^b	0.82±0.05 ^b	0.32±0.03 ^a	0.56±0.20 ^b	0.78±0.12 ^b	0.23±0.03 ^a	0.44±0.22 ^b	0.53±0.23 ^b	0.12±0.03 ^a	0.33±0.07 ^b	0.21±0.01 ^b	0.22±0.08 ^a	0.53±0.11 ^b	0.45±0.03 ^b	0.56±0.09 ^a	1.21±0.38 ^b	1.5
0.44±0.02 ^a	0.78±0.2 ^b	0.87±0.03 ^b	0.12±0.03 ^a	0.44±0.22 ^b	0.36±0.13 ^b	0.13±0.05 ^a	0.30±0.09 ^b	0.34±0.04 ^b	0.30±0.13 ^a	0.71±0.17 ^b	0.67±0.10 ^b	0.21±0.12 ^a	0.42±0.15 ^b	0.55±0.14 ^b	0.09±0.01 ^a	0.19±0.18 ^b	0.16±0.04 ^b	0.23±0.04 ^a	0.42±0.20 ^b	0.3
4.53±0.37 ^a	10.65±1.04 ^b	8.75±0.63 ^b	7.04±0.32 ^a	12.03±0.73 ^b	11.53±1.31 ^b	3.22±0.54 ^a	8.80±0.57 ^b	9.06±1.40 ^b	5.52±0.05 ^a	10.93±0.53 ^b	8.97±1.02 ^b	2.39±0.61 ^a	4.23±0.15 ^b	5.43±0.12 ^b	3.56±0.07 ^a	6.97±0.39 ^b	9.65±1.22 ^c	3.49±0.62 ^a	6.56±0.09 ^b	5.5
13.56±0.86 ^a	5.43±0.84 ^b	5.49±0.20 ^b	18.00±0.61 ^a	7.56±0.31 ^b	7.01±0.81 ^b	10.15±0.27 ^a	5.85±0.28 ^b	4.34±0.50 ^b	11.67±2.30 ^a	6.66±0.52 ^b	6.91±0.51 ^b	7.51±0.85 ^a	4.87±0.29 ^b	3.85±0.50 ^b	8.94±1.04 ^a	4.78±1.43 ^b	4.32±0.80 ^b	9.10±0.46 ^a	3.87±1.98 ^b	4.8
5.18±0.49 ^a	11.85±1.58 ^b	9.95±0.71 ^b	7.48±0.41 ^a	13.24±1.31 ^b	12.71±1.49 ^b	3.67±0.62 ^a	9.66±0.86 ^b	10.18±1.56 ^b	6.05±0.21 ^a	12.08±0.92 ^b	10.17±1.35 ^b	2.72±0.76 ^a	4.98±0.37 ^b	6.19±0.27 ^c	3.87±0.16 ^a	7.69±0.68 ^b	10.26±1.29 ^c	4.28±0.75 ^a	8.19±0.67 ^b	7.4

WT: wild-type. Data are expressed as means ± S.D.; different superscript letters (a-c) represent statistical significant differences ($P < 0.05$).

Table S4. PUFA compositions of seven major tissues in *Fad2-Fad3* F0/F1 transgenic mice and wild-type mice.

Skeletal muscle			Fat			Heart			Liver			Spleen			Lung			Kidney		
WT (n=8)	Fad2-Fad3 F0(n=5)	Fad2-Fad3 F1(n=8)	WT (n=8)	Fad2-Fad3 F0(n=5)	Fad2-Fad3 F1(n=8)	WT (n=8)	Fad2-Fad3 F0(n=5)	Fad2-Fad3 F1(n=8)	WT (n=8)	Fad2-Fad3 F0(n=5)	Fad2-Fad3 F1(n=8)	WT (n=8)	Fad2-Fad3 F0(n=5)	Fad2-Fad3 F1(n=8)	WT (n=8)	Fad2-Fad3 F0(n=5)	Fad2-Fad3 F1(n=8)	WT (n=8)	Fad2-Fad3 F0(n=5)	Fad2-Fad3 F1(n=8)
24.32±2.99 ^a	12.32±1.03 ^b	13.02±1.26 ^b	28.01±3.14 ^a	10.03±1.06 ^c	12.21±1.73 ^b	16.66±1.26 ^a	10.44±1.73 ^b	8.78±0.34 ^c	17.78±1.23 ^a	8.47±2.03 ^b	8.34±0.43 ^b	13.32±0.37 ^a	6.54±2.07 ^b	7.75±2.54 ^b	12.21±1.14 ^a	4.33±0.41 ^b	3.23±0.02 ^b	14.03±1.4 ^a	10.04±0.56 ^b	8.78±0.34 ^c
12.03±0.42 ^a	15.01±1.17 ^b	15.65±0.21 ^b	15.45±0.09 ^a	19.21±1.9 ^b	18.31±1.05 ^b	8.74±0.04 ^a	13.74±2.05 ^b	12.03±1.74 ^b	10.02±0.38 ^a	14.01±2.03 ^b	15.04±1.03 ^b	6.54±0.49 ^a	9.98±1.32 ^b	9.34±0.55 ^b	6.56±0.73 ^a	10.22±0.89 ^b	9.49±1.30 ^b	7.84±0.31 ^a	12.75±1.48 ^b	11.03±0.31 ^b
0.32±0.02 ^a	1.21±0.05 ^b	1.53±0.03 ^c	0.24±0.02 ^a	0.41±0.38 ^b	0.44±0.12 ^b	0.23±0.01 ^a	0.60±0.41 ^b	0.41±0.03 ^b	0.32±0.02 ^a	0.65±0.21 ^b	0.43±0.12 ^b	0.07±0.04 ^a	0.12±0.08 ^a	0.23±0.05 ^b	0.07±0.02 ^a	0.13±0.01 ^a	0.14±0.01 ^b	0.17±0.02 ^a	0.34±0.09 ^b	0.31±0.02 ^b
1.21±0.42 ^a	2.21±0.71 ^b	1.56±0.22 ^b	2.31±0.50 ^a	3.31±0.04 ^b	3.27±0.38 ^b	1.18±0.22 ^a	2.54±0.06 ^b	2.12±0.13 ^b	1.33±0.40 ^a	2.98±0.69 ^b	2.74±0.22 ^b	0.90±0.32 ^a	1.93±0.92 ^b	2.53±0.38 ^b	2.31±0.29 ^a	3.67±0.09 ^b	3.33±0.82 ^b	1.09±0.13 ^a	1.71±0.19 ^b	1.71±0.19 ^b
0.21±0.10 ^a	1.21±0.14 ^b	0.89±0.05 ^b	0.32±0.06 ^a	0.61±0.31 ^b	0.72±0.20 ^b	0.32±0.03 ^a	0.55±0.03 ^b	0.45±0.04 ^b	0.23±0.03 ^a	1.12±0.46 ^b	1.12±0.03 ^b	0.12±0.03 ^a	0.24±0.16 ^b	0.22±0.03 ^b	0.22±0.08 ^a	0.64±0.03 ^b	0.89±0.12 ^b	0.56±0.09 ^a	1.98±0.13 ^b	1.98±0.13 ^b
0.44±0.02 ^a	2.22±0.03 ^b	1.77±0.23 ^b	0.12±0.03 ^a	0.54±0.08 ^b	0.63±0.09 ^b	0.13±0.05 ^a	0.54±0.16 ^b	0.42±0.13 ^b	0.30±0.13 ^a	1.56±0.21 ^b	1.67±0.21 ^b	0.21±0.12 ^a	0.43±0.22 ^b	2.53±0.40 ^c	0.09±0.01 ^a	0.47±0.06 ^b	0.45±0.05 ^b	0.23±0.04 ^a	0.63±0.09 ^b	0.71±0.02 ^b
4.53±0.37 ^a	17.34±1.21 ^b	16.23±1.55 ^b	7.04±0.32 ^a	19.32±0.98 ^b	21.43±1.61 ^b	3.22±0.54 ^a	15.54±0.05 ^c	11.44±2.01 ^b	5.52±0.05 ^a	15.96±0.22 ^b	17.43±1.76 ^b	2.39±0.61 ^a	8.87±0.52 ^b	9.49±0.29 ^b	3.56±0.07 ^a	12.98±1.07 ^b	10.79±0.49 ^b	3.49±0.62 ^a	12.94±1.08 ^b	15.03±0.31 ^b
13.56±0.86 ^a	18.43±1.93 ^b	18.74±0.46 ^b	18.00±0.61 ^a	22.93±2.32 ^b	22.02±1.55 ^b	10.15±0.27 ^a	16.88±2.52 ^b	14.56±1.90 ^b	11.67±2.30 ^a	17.64±2.93 ^b	18.21±1.37 ^b	7.51±0.85 ^a	12.03±2.32 ^b	12.10±0.98 ^b	8.94±1.04 ^a	14.02±0.99 ^b	12.96±2.13 ^b	9.10±0.46 ^a	14.8±1.76 ^b	13.56±0.86 ^a
5.18±0.49 ^a	20.77±1.38 ^b	18.89±1.83 ^b	7.48±0.41 ^a	20.47±1.37 ^b	22.78±1.90 ^b	3.67±0.62 ^a	16.63±0.24 ^c	12.31±2.18 ^b	6.05±0.21 ^a	18.64±0.89 ^b	20.22±2.00 ^b	2.72±0.76 ^a	9.54±0.90 ^b	12.24±0.72 ^c	3.87±0.16 ^a	14.09±1.16 ^b	12.13±0.66 ^b	4.28±0.75 ^a	15.55±1.30 ^b	18.89±1.83 ^b

WT: wild-type. Data are expressed as means ± S.D.; different superscript letters (a-c) represent statistical significant differences ($P < 0.05$).

Comment 6:

Figure 1e, 2e, 3d and 3e: The results of PUFA analysis in Figure 3d and 3e have error bars, but not shown in Figure 1e and 2e. The authors should not omit this.

Response 6:

We thank the reviewer for bringing this to our attention. We re-analysed the data of Figure 1~4, and relabeled the significance superscript. Moreover, we added the source data in the “Supplementary Material” section (response Figure 2B; 2E-F; revised manuscript electronic supplementary material, table S3-4).

Comment 7:

What is the difference in the results of PUFA analysis between F0 and F1 transgenic mice? The authors need to address this critical point.

Response 7:

Thank you for raising this point. In accordance with your suggestion, we have added new PUFA analysis experiments to the revised manuscript.

➤ For the *Fad3* single transgenic mice:

(revised manuscript page 9, line 190-196) “Moreover, the ratio of n-6/n-3 PUFAs in the F1 transgenic mice was similar to the ratio in F0 (skeletal muscle: from 2.62 to 0.46, $P < 0.05$; fat: from 2.41 to 0.57, $P < 0.05$; heart: from 2.77 to 0.61, $P < 0.05$; liver: from 1.93 to 0.55, $P < 0.05$; spleen: from 2.76 to 0.98, $P < 0.05$; lung: from 2.31 to 0.62, $P < 0.05$; and kidney: from 2.13 to 0.47, $P < 0.05$; response Figure 2C; revised manuscript figure 2c). These results indicate that the plant *Fad3* gene can be functionally expressed in the major tissues of F0 and F1 transgenic mice and *Fad3* play an active role in conversion of n-6 into n-3 PUFAs.”

➤ For the *Fad2-Fad3* double transgenic mice:

1. (revised manuscript page 11, line 234-239) “To assess the function of the *Fad2-Fad3* enzymes in the F0 and F1 transgenic mouse, the PUFA content in the major tissues were compared, including skeletal muscle, heart, liver, spleen, lung, kidney, and fat. response Figure 2E; revised manuscript figure 4b displays the PUFAs profile of seven tissues from F0 and F1 of the *Fad2-Fad3* mouse. It shows that all the n-6 and n-3 PUFAs were higher in compared with wild-type mice, indicating that the *Fad2-Fad3* double transgenic mice had efficiently converted the monounsaturated fatty acids into n-6 and n-3 PUFAs in their bodies.”

2. (revised manuscript page 12, line 253-258) “The seven major tissues from *Fad2-Fad3* double transgenic mice were collected and analyzed for n-6/n-3 PUFAs ratio. Notably, F0 and F1 double transgenic mice showed a substantially lower n-6/n-3 ratio in all tissues examined compared with wild-type mice (skeletal muscle: 0.99, 0.89 vs. 2.61; fat: 1.12, 0.97 vs. 2.41; heart: 1.18, 1.02 vs. 2.77; liver: 0.90, 0.95 vs. 1.93; spleen: 0.98, 1.26 vs. 2.76; lung: 1.07, 1.00 vs. 2.31; kidney: 0.72, 0.95 vs. 2.13; response Figure 3A, below; revised manuscript figure 5b).”

Figure 3 Response to Reviewer

(A) The ratio of n-6/n-3 PUFAs in wild-type and *Fad2-Fad3* double transgenic mice (different superscript letters (a-c) in each column represent statistical significant differences, $P < 0.05$; the bars represent mean \pm S.D.).

Comment 8:

There are a few grammar and word mistakes that should be corrected. We recommend that the manuscript is edited by a native speaker, to improve its flow and clarity.

Response 8:

English is not our native language, we believe that the major critical issue for our manuscript is the poor preparation of the manuscript. According your suggestion, the revised manuscript has been checked by a native English speaker to ensure the language quality of our manuscript.